# Control System of a Motor-Driven Precision No-Tillage Maize Planter Based on the CANopen Protocol

**Jincheng Chen [1], Hui Zhang [1], Feng Pan [1], Mujun Du [2] and Chao Ji [1,*]**

[1] Institute of Mechanical Equipment, Xinjiang Academy of Agricultural and Reclamation Science, Shihezi 832003, China; shznkycjc@163.com (J.C.); zhanghxj2021@163.com (H.Z.); 17609922366@163.com (F.P.)

[2] Heilongjiang dewo Science & Technology Development Co., Ltd., Harbin 150030, China; dumujun@163.com

* Correspondence: jicobear@163.com

**Abstract:** To reduce the cost of machinery and manual operation, greatly improve the efficiency of maize sowing, and solve the problems of slow sowing speed, unstable operation quality, and the difficult monitoring of the sowing process of traditional seeders, a control system for an electrically driven precision maize seeder based on the CANopen protocol was designed. In this system, an STM32 is used as the main controller, and the vehicle terminal is used to set the operating parameters, such as the spacing of sowing plants and the number of holes in the metering plate. The GPS receiver is used to collect the forward speed of the tractor. An infrared photoelectric sensor is used to monitor the working state of the seeder. In this study, tests were conducted on different evaluation indices. The results showed that the detection accuracy of the photoelectric sensor reached 99.8% and the fault alarm rate reached 100%. The qualified rate of sowing was more than 91.0%. Based on indoor test results, the qualified rate was higher when the grain spacing was larger. The field test showed, in terms of the seeding performance, that the control system had good stability. When the grain spacing was set to 20 cm and the operating speed was 6~12 km/h, the qualified index was more than 89% and the reseeding index was less than 1.93%. The variation in sowing performance between different monomers was small, and the seeding performance was good. The control system helps to improve the performance of the seeder.

**Keywords:** precision planter; motor-driven; CANopen protocol; photoelectric sensor; no-tillage





## 1. Introduction

Maize is the largest food crop in China and occupies an extremely important position in the whole agricultural planting system. In the new era, with the rapid development of China's economy, the actual demand for corn has increased greatly [1,2]. Changes in corn supply and demand have a great influence on maintaining national food security and stabilizing the grain market and supply [3]. In recent years, with the major management mode of agricultural production gradually developing to a large-scale and intensive direction, to effectively ensure the cultivated area and grain production task and the completion of sowing operations in high-yield periods, higher quality requirements have been proposed for maize precision sowing technology [4–6]. Facing the higher cost-savings and efficiency requirements of farmers and the more urgent demand for agricultural time, the deficiencies of ground wheel drive are increasingly prominent [7,8]: (1) Low operation efficiency. At present, the operation of precision planters in China is still at a low-speed level of 6~8 km/h. (2) Unstable sowing quality. Under high-speed operation, it is easy to bump and slip, resulting in a series of problems, such as missed sowing, reseeding, and poor uniformity of plant spacing. (3) It is hard to monitor the sowing process [9]. Traditional machines and tools operate in a closed environment, requiring auxiliary personnel to follow the machine and observe, which is not only labor-intensive and costly but also easily causes personal injury, and the observation results make it difficult to eliminate the influence

of human subjective factors. Sensor-based electronic metering systems can minimize the lacunae of mechanical metering systems. The application of electronic seed metering and control systems in planters is required for better seed uniformity in the field [10].

Sound and sustainable agriculture without electronics is inconceivable today, as electronic systems are used to reduce farm inputs, protect the environment, secure farm income, and produce high-quality products [11]. In the last few decades, a number of active seeding control and detection systems have been proposed to solve the above-mentioned problems. Yuan et al. [6] used prescription operation maps and GPS information, combined with speed, to drive a servo motor seed space and achieved precision planting that could be steplessly adjusted from 10 to 20 cm. Yang et al. [12] designed a mechatronic driving system. Compared to the mechanical driving system, the advantage of the mechatronic driving system is noticeable, especially when the forward speed is more than 11 km/h. Anil et al. [13,14] developed an electromechanical drive system (EMDS) for seed metering units of a classic single-seed planter to attain uniform seed spacing. EMDS realizes the recommended optimal seeding rate; the possibility for fast and simple setting, synchronization, and real-time control; the ability to work at higher speeds; single movement; and the control of each metering unit. The dynamic relationship model between the speed of the tractor and the speed of the metering plate is established to ensure the accurate matching of the tractor time and the seed entry to better realize seed spacing consistency. Ding et al. [15] proposed a control system of a motor-driven precision maize planter based on GPS speed measurements. At the same plant spacing and operating speed, the variation coefficient of the GPS velocity measurement method is smaller than that of the encoder velocity measurement method. At a high speed of 12 km/h, the average qualified GPS index is 14.32% higher than that of the encoder. This shows that the GPS velocity measurement method is more suitable for high-speed operation. Li et al. [16] resolved the problem that GPS receivers cannot meet the requirement of precision seeding at low speed based on a Kalman filter.

Variable-rate seeding (VRS) technology can adjust the seed input according to regional soil differences, ensure the most suitable plant density, make full use of nutrients and moisture in the soil, and exert the maximum yield potential in specific soil regions, thus significantly increasing yield and reducing cost. He et al. [17,18] developed a low-cost VRS control system based on a controller area network (CAN) bus and developed a compensation algorithm for seeding lag (CASL) that could decrease the seeding lag distance immensely. The developed VRS control system was capable of flexibly expanding planter rows and independently controlling each row's seeding rate. Ding et al. [19] developed a variable rate planter row-unit driver for maize. The overall test results of the row-unit driver confirmed that it could realize the functions of seed metering, seeding quality detection, and CAN communication with the main controller.

To improve the seeding uniformity of a maize planter, He et al. [20] designed a GPS-based turn compensation algorithm to offset the seeding rates of planter units. Field experiments indicated that a four-row planter equipped with the developed turn compensation control system had seeding accuracies (above 97%) and seeding coefficients of variation (below 1.52%) values better than those of a noncompensation planter under equivalent working conditions. To find the problem of seeding blockage and missing seeding in time, Meng et al. [21] developed a monitoring system to solve the phenomenon of maize precision seeding machines in operation and to improve the economy and efficiency of seeding. Xie et al. [22] conducted a study testing the accuracy of the sensor to monitor the seeding parameters of a precision metering device under different seeding speeds and seeding spacings. Improving the accuracy of the sensor's monitoring of the seed passing frequency is of great help in improving the seeding monitoring accuracy under the conditions of high seeding speed and small seeding spacing. Xie et al. [7] developed a precision seeding parameter monitoring system based on laser sensors Field tests showed that the average monitoring error of the seeding quantity was less than 1%, and the average

monitoring error of the seeding qualified rate was less than 1.5%. The monitoring system could trigger an alarm in time when the seeder had a missing seed fault.

In summary, most field experiments involving the seeder use four-row or six-row mechanical seeders. For the eighteen-row air suction seeder, in this study, an electrically driven precision sowing control system based on the CANopen protocol was designed, and a circuit board integrating the motor drive and sowing quality detection was developed. A seeding parameter dictionary with the CANopen protocol was constructed. A separable trapezoidal integral proportional integral derivative (PID) control algorithm was used to match the tractor speed and motor speed. In this paper, the performance of the control system was evaluated by laboratory bench and field tests.

## 2. Materials and Methods

### 2.1. System Components

The proposed maize precision planter system consisted of a monitoring subsystem and a mechanical device system. As shown in Figure 1, the required hardware components included a 12 V DC power supply, an on-board computer with a CAN bus (eMT3070B, Weintek Technology Co., Ltd., New Taipei, China), an in-house-designed integrated controller based on STM32F103VET6, an infrared monitoring sensor (Shandong Zhucheng Dilico Automotive Electronics Co., Ltd., Weifang, China), an inertial and satellite navigation module (WTGPS-200 WitMotion Shenzhen Co., Ltd., Shenzhen, China), brushed DC motors, and in-house-designed motor speed measurement modules. The mechanical part included a reducer, a planter plate, and a seed tube. The motor was used as an intermediary to integrate the control system and mechanical part.

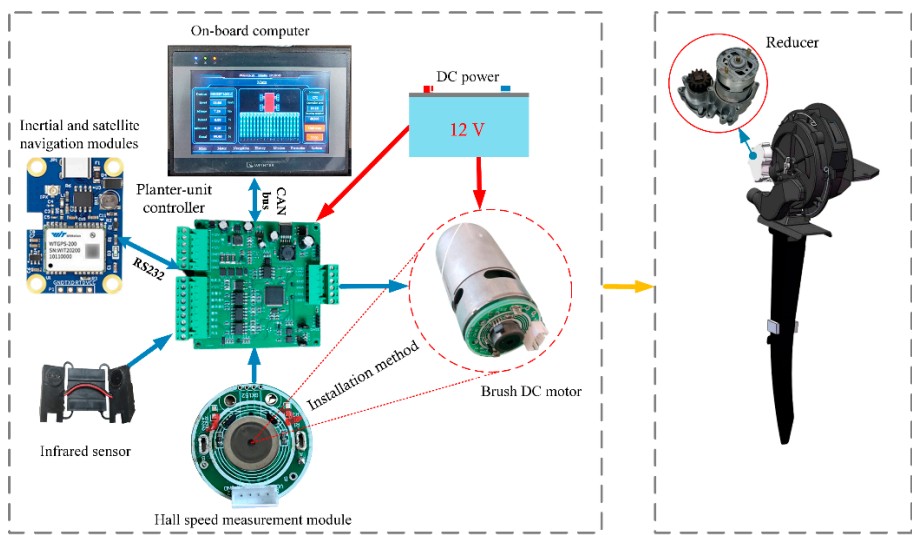

**Figure 1.** Planter monitoring system.

In detail, the on-board computer communicated with the controller via a CAN bus and was used for setting the seed spacing, current threshold, and width; monitoring the various working states of the system (such as the motor current and rotational speed); and controlling the start and stop of a single motor. To reduce field wiring, in this paper, the controller was integrated with the motor drive and CAN communication, which was mounted on each planter unit, to expand flexibly based on the planter row number and to adjust the motor speed to achieve the desired seeding rate. In this study, speed acquisition was performed through inertial and satellite navigation modules with a velocity accuracy of 0.05 m/s, and bidirectional credit guaranteed communication with the controller through RS232. A brush DC motor was utilized to drive the seed meter at a desired speed, and a hall speed measurement module for the brushless DC motor was developed, nested on the shaft side of the motor, and a pulse was generated by an interaction with the magnetic

ring on the motor shaft. Additionally, a photoelectric sensor with a large field of view, high sensitivity, and strong dust resistance was installed on the seed tube to monitor the state of falling seeds. A circuit schematic diagram of the system with STM32F103VET6 as the main controller is shown in Figure 2.

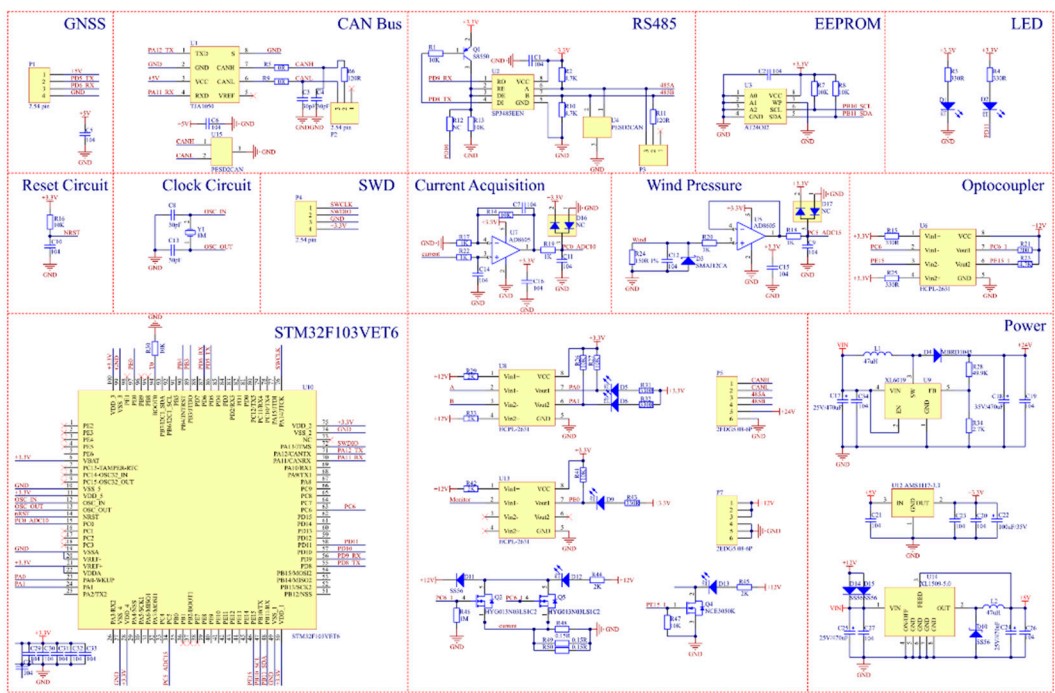

**Figure 2.** Schematic diagram of the system circuit.

## 2.2. CAN Bus with CANopen Protocol

The exchange of data packets in the system was based on a CAN bus. However, ISO 11,783 was specifically designed for tractor control system development [23–25]. In China, fewer products have been developed using subprotocols, especially in sensors and on-board computers. The on-board computer (eMT3070B) used in this paper had a CAN interface and conformed to the CANopen protocol. To test the designed system as soon as possible and to enhance the scalability at the present stage, the CANopen protocol was used as the basic protocol. CANopen is a high-level communication protocol based on the controller area network. It includes a communication subprotocol and a device subprotocol and has often been presented in embedded systems and industrial controls [26]. The CANopen protocol usually consists of three parts: a user application layer, an object dictionary, and communication. The core part is the object dictionary, which describes the relationship between the application object and the CANopen message. The user application layer in this paper refers to the application interface downloaded to the eMT3070B using EasyBuilder Pro development software provided by Weintek. Figure 3 shows the partial display interface design of the monitoring software for the eighteen-row seeding. In the communication layer, considering the field working environment, the well-established TJA1050 chip was selected as the transceiver of the CAN bus. This chip can work normally even with electrostatic interference and in voltage-mutating and high-noise environments and communicates with electricity.

To be stable, reliable, and controllable, the CANopen network needs to be set up with a network management master (NMT-Master) that controls the start and stop of all nodes. Communication between the on-board computer as the NMT host and the NMT slave via the NMT network management message is responsible for the layer management, network management, and ID distribution services. NMT management involves six states of a CANopen node following power-up: initializing, application reset, communication reset,

preoperational, operational, and stopped. The NMT management state transition diagram is shown in Figure 4.

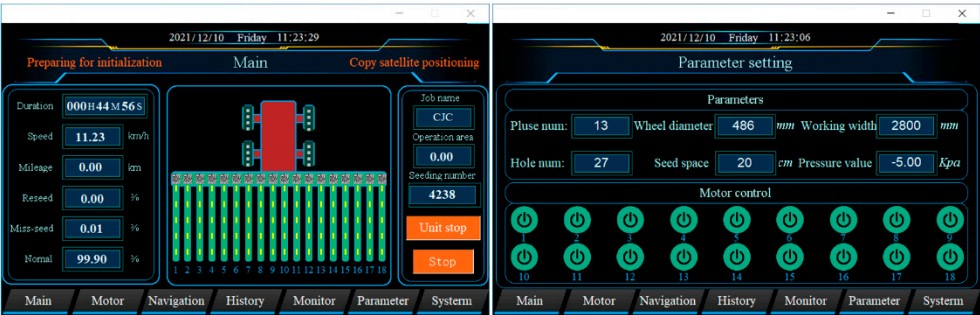

**Figure 3.** Partial human–computer interaction software interface.

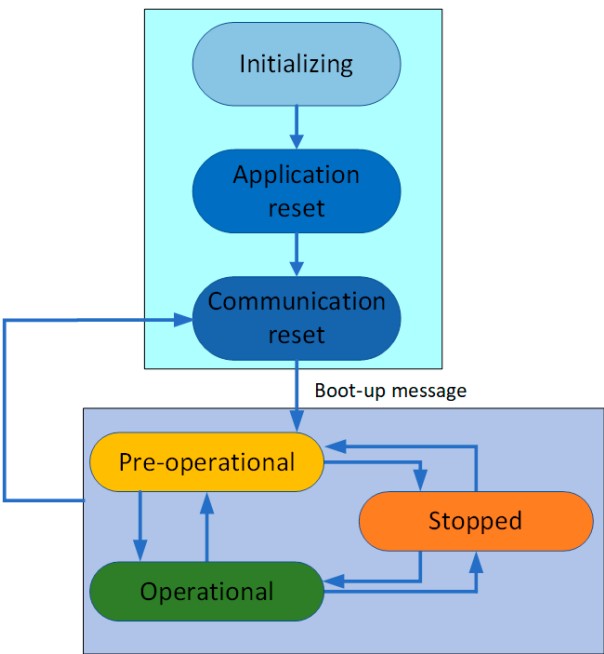

**Figure 4.** NMT management state transition diagram.

The object dictionary is the most important part of a device specification. It is an ordered set of parameters and variables, including all parameters of device description and device network state. The CANopen protocol uses an object dictionary with 16-bit indices and 8-bit subindices, and all parameters of the device can be accessed through the object dictionary. The parameter object dictionary of the system is defined in the 2000H–5000H (H represents hexadecimal system) index region according to the CANopen CiA 301 document. Real-time data use the process data object (PDO) for asynchronous one-way transmission without a node response. The service data object (SDO) is mainly used for the parameter configuration of slave nodes in the CANopen master station. Service validation is the largest feature of an SDO, generating a response for each message to ensure the accuracy of data transmission. The CAN bus system in this paper consisted of a master node and eighteen slave nodes with 104 object dictionaries. Partial object dictionary descriptions are shown in Table 1. The CAN bus data transmission mode is shown in Figure 5.

**Table 1.** Partial object dictionary description.

| Parameter | Number of Bits | Transport Types | Dictionary Index | |
|---|---|---|---|---|
| | | | Indices | Subindices |
| Operating speed | 32 | PDO | 2000 | 00 |
| Operating area | 32 | PDO | 2003 | 00 |
| Motor status | 8 | PDO | 2004 | 00 |
| Seeding status | 8 | PDO | 2005 | 00 |
| Seeding number | 32 | PDO | 2006 | 00 |
| Miss-seeding rate | 32 | PDO | 2007 | 00 |
| Replay rate | 32 | PDO | 2008 | 00 |
| Seed spacing | 16 | SDO | 2001 | 00 |
| Motor control | 8 | SDO | 2009 | 00 |
| Working width | 16 | SDO | 200A | 00 |
| ... | ... | ... | ... | ... |

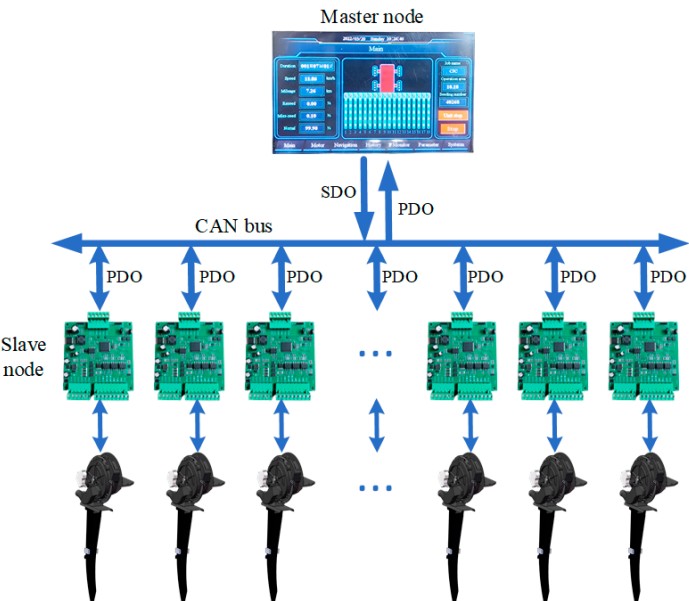

**Figure 5.** CAN bus data transmission mode.

### 2.3. Motor Speed Matching Operation Speed

To ensure the uniformity and qualified rate of seed spacing, it is very important to establish the dynamic matching relationship between the motor speed and the tractor speed. The rotational speed of the motor is determined by the tractor speed, the number of holes in the seeding plate, the transmission ratio from the reducer to the seeding plate, and the seeding distance. Accordingly, the required rotary speed of the planter unit can be calculated as:

$$R = \frac{1000 V I}{36 \, X_{ref} N} \tag{1}$$

where $R$ is the motor speed (r/s), $V$ is the tractor speed (km/h), $I$ is the transmission ratio from the reducer to the seeding plate, $X_{ref}$ is the setting seeding distance (cm), and $N$ is the number of holes in the seeding plate. For a well-processed seeding cell, $I$ and $N$ are fixed values. $X_{ref}$ is set based on the agricultural technology. Therefore, the tractor speed is the most critical factor affecting the sowing quality.

### 2.4. Speed Acquisition and Motor Control

GPS speed measurement is not affected by the structure of the seeder and surface conditions and can provide a variety of data, including latitude and longitude, heading

angle, and elevation. Compared with other velocity measurement methods, such as encoders, it has great advantages. WTGPS-200 is a high-performance vehicle-mounted integrated navigation system for vehicle navigation. When the signal accuracy of the GNSS system is reduced or if the satellite signal is lost, the WTGPS-200 system uses pure inertial navigation technology without the aid of odometer information. It can also independently carry out high-precision positioning, velocity measurement, and attitude measurement for vehicle carriers over a long time. The accuracy of 0.05 m/s can meet the requirements of the GBT6973-2005 single-seed (precision) seeder test method. The controller obtains GPRMC frames conforming to the NMEA0183 protocol by RS232. Figure 6 shows the GPRMC frame format with fifteen fields. Field 0, as the frame head, represents the beginning of a frame, field thirteen is the frame data validation, and the frame ends with CR/LF. Field one to field twelve represent the data fields, in which field seven represents the speed value. Therefore, the seventh field in a frame can be extracted to obtain the speed.

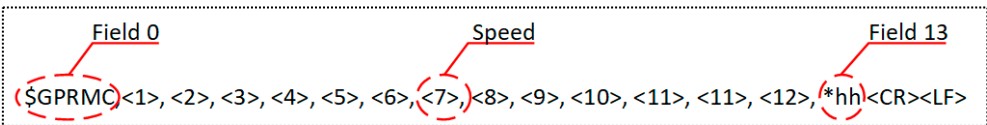

**Figure 6.** GPRMC frame format.

The real-time motor speed was controlled via the pulse width modulation (PWM) signal generated by the STM32 chip's internal timer. The PWM mode could generate a signal whose frequency was determined by the TIMx_ARR register, and the duty ratio was determined by the TIMx_CCRx register. The duty ratio could be adjusted to control the motor speed at a certain PWM frequency. Limited to the computing power of the chip used, more complex intelligent control algorithms are not adopted, such as adaptive PID [27], particle swarm optimization algorithm [28], fuzzy PID Control Algorithm [29,30], and ant colony optimization [31]. On the other hand, the experimental results indicated that the motor speed showed a linear relationship with the duty ratio. Therefore, closed-loop control can be carried out by the PID control algorithm [32]. PID control is a closed-loop control method based on deviation, which can eliminate the deviation between the target speed and the actual speed of the motor in the adjustment process. In discrete PID control, the realization of integration is the rectangular addition calculation in the case of infinite subdivision. In the discrete state, the time interval is large enough, and the accuracy of rectangular integration appears to be lower in some cases. To minimize the difference, the rectangular integration was changed into trapezoidal integration to improve the calculation accuracy. Introducing the trapezoidal integral into the incremental PID algorithm modifies the formula as follows:

$$\Delta v(k) = K_p(e(k) - e(k-1)) + K_i \frac{e(k) + e(k-1)}{2} + K_d(e(k) - 2e(k-1) + e(k-2)) \quad (2)$$

where $\Delta v(k)$ is the adjustment value, $K_p$ is the proportional coefficient, $K_i$ is the integral coefficient, $K_d$ is the differential coefficient, and $e(k-1)$, $e(k)$, and $e(k-2)$ are the last three deviations. Figure 7 shows an analysis of the bench test data. The optimum motor speed control could be achieved when $K_p$ was 4.15, $K_i$ was 1.2, and $K_d$ was 0.

The theoretical motor speed calculated by Formula (1) is the target value; the rotor position sensor measures the speed signal as a feedback value. The theoretical calculation of the target speed does not consider the influence of external factors. However, due to the factors of actual operation, such as zero drift of the speed sensor, error of DC motor speed measurement, and the efficiency of mechanical transmission, the error of the control parameters ($e(k)$) is affected. Therefore, setting a threshold variable, t, does not perform the PID algorithm when the deviation is less than the absolute value of the threshold. Experimental results showed that the control precision was best when the absolute value of threshold t was 0.15. On the other hand, if a system always has a uniform direction

deviation, infinite accumulation and saturation can occur, which greatly affects the system performance. To solve the problem of integral saturation, the PID algorithm anti-integral saturation was introduced. The idea is to determine whether the control, $C(k-1)$, of the previous moment has exceeded the limit when calculating $e(k)$. If $C(k-1) > C_{max}$ ($C_{max}$: sets the TIMx capture compared to the register maximum value), only negative deviations are accumulated; if $C(k-1) < C_{min}$ ($C_{min}$: sets the TIMx capture compared to the register minimum value), only positive deviations are accumulated. This avoids the control quantity from staying in the saturated zone for a long time. The PID control algorithm is shown in Figure 8.

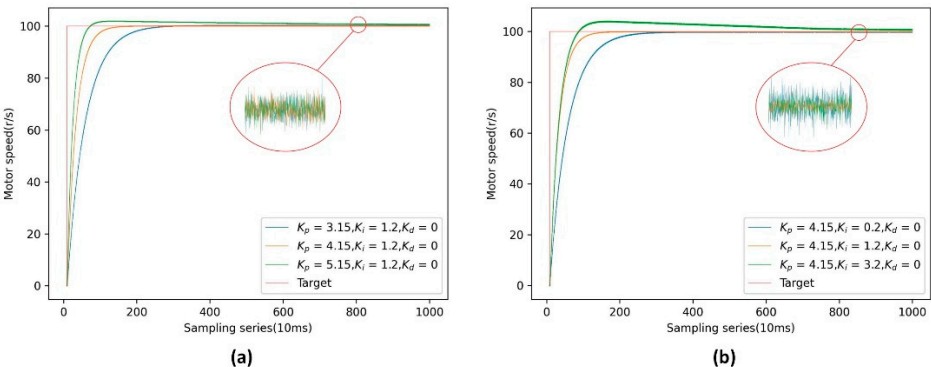

**Figure 7.** Data analysis curve of different PID parameters: (**a**) response curves under different $K_p$ conditions and (**b**) response curves under different $K_i$ conditions.

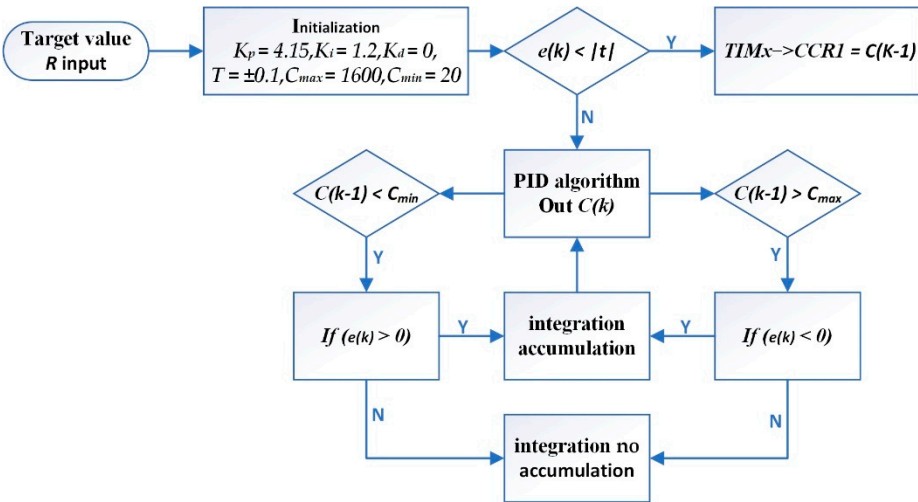

**Figure 8.** PID control algorithm.

### 2.5. Sowing Monitoring

To realize the real-time monitoring of the quality of maize no-tillage precision seeding operations, a seeding monitoring system based on reflective infrared photoelectric induction was designed. The monitoring probe used an infrared emitting diode and a photodiode as the signal transmitting and receiving ends. During the seeding operation, corn seeds were separated into single seeds from the seed metering device, dropped into the seed guiding tube, and were finally discharged into the soil through the lower seed guiding mouth. Among the working components involved in the seeding process, the structure of the seed guiding tube was the simplest and the closest to the seed dropping point. Therefore, mounting the seed monitoring probe on the seed guiding tube was preferred.

According to GB/T 6973-2005, the ratio of actual adjacent seed spacing, $X$ (cm), to theoretical seed spacing, $X_{ref}$ (cm), is the benchmark for evaluating the quality of seed metering. In addition to field measurements, the actual seed spacing is generally estimated

by multiplying the tractor speed, $V$ (km/h), of the seeder by the interval time, $T$ (ms), between adjacent seeds. The forward speed, $V$ (km/h), of the seeder can be obtained by the pick-up circuit. Therefore, the comparison between the actual seed spacing and the theoretical seed spacing can be converted to a numerical comparison between the actual adjacent seed falling time interval, $T$ (ms), and the theoretical time interval, $T_0$ (ms). According to the standard, if $X > 1.5X_{ref}$, it is judged as a miss-seeding, and if $X \leq 0.5X_{ref}$, the seeding is judged as a reseed. For the convenience of system calculation, the judgment basis is converted to the relationship between the tractor speed, $V$ (km/h), and the theoretical distance, $X_{ref}$ (cm). If $VT > 54X_{ref}$, the seeding is judged as a miss-seeding. If $VT \leq 18X_{ref}$ is judged as a reseeding and if $18X_{ref} < VT < 54X_{ref}$, the seeding is a quality seeding. When a fault (miss-seeding or reseed) occurs, an alarm is triggered. Figure 9 shows three different states of falling seeds in the seed tube. Figure 10 shows the seed condition monitoring process.

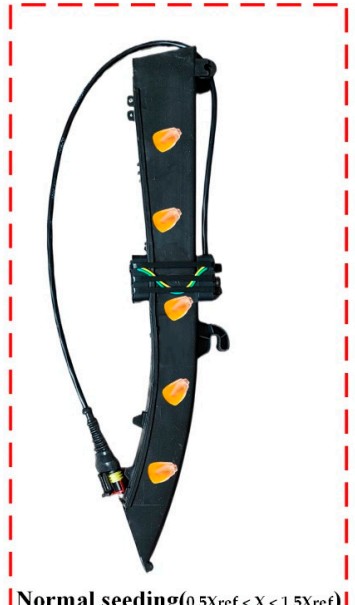 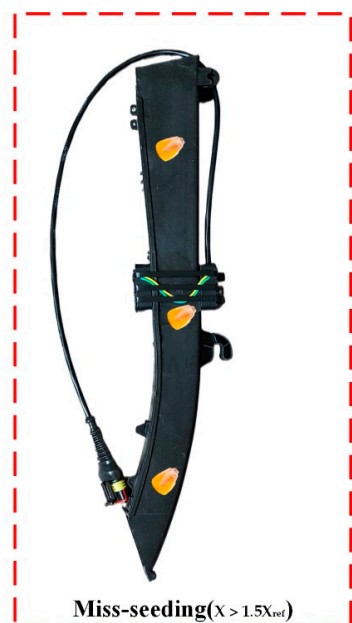 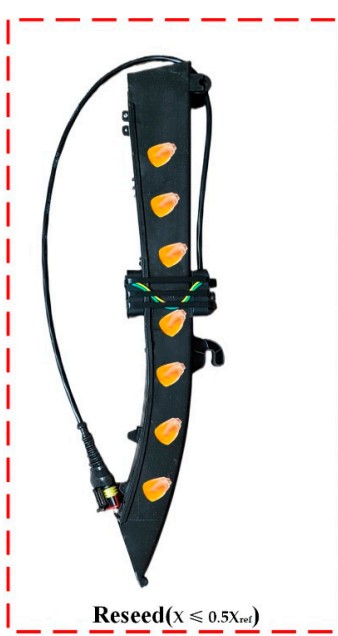

**Normal seeding**($0.5X_{ref} < X < 1.5X_{ref}$)   **Miss-seeding**($X > 1.5X_{ref}$)   **Reseed**($X \leq 0.5X_{ref}$)

**Figure 9.** Judging the state of falling seeds in the seed tube.

### 2.6. Performance Test of the Seeder Monitoring and Control System

To verify the performance of the seeder monitoring and control system, laboratory bench tests and field tests were conducted. These tests included photoelectric sensor detection performance tests, abnormal alarm rate reliability tests, motor dynamic speed response tests, and statistical analyses of real-time sowing monitoring parameters.

The related tests were carried out on the JPS-12 seed metering device performance test bench (Bona Technology Co., Ltd., Harbin, China). The test materials were Xinyu No. 9 hybrid maize seeds produced by the Crop Research Institute of Xinjiang Academy of Agricultural Reclamation Sciences. The moisture content was 9.10%, the purity was 98.75%, and the thousand-grain weight was (274.22 ± 2.52) g. We randomly measured 300 seeds, and the shape was horse tooth, and the length, width, and height were 10.04 ± 1.06 mm, 7.45 ± 0.86 mm, and 5.50 ± 1.01 mm, respectively.

The seeding unit motor drive control system and experimental test setup are shown in Figure 11. The metering device was an air suction seed metering device produced by Precision Planting Company in the United States. The diameter of the metering plate was 4.5 mm, and the number of seed holes was 27. The DC motor was an NC3SFN-6035-CVC carbon brush variable-resistance brush DC motor produced by Transmotec, Sweden. The working voltage was 12 V, the current was 5.6 A, the rated speed was 10,700 r/min, and the stall torque was 446.8 mN·m. The motor reducer was a three-stage gear reducer developed

by Devo, Heilongjiang Province, and the deceleration ratio was 82.8125. The power output gear of the DC motor reducer engaged with the external gear of the seeding plate.

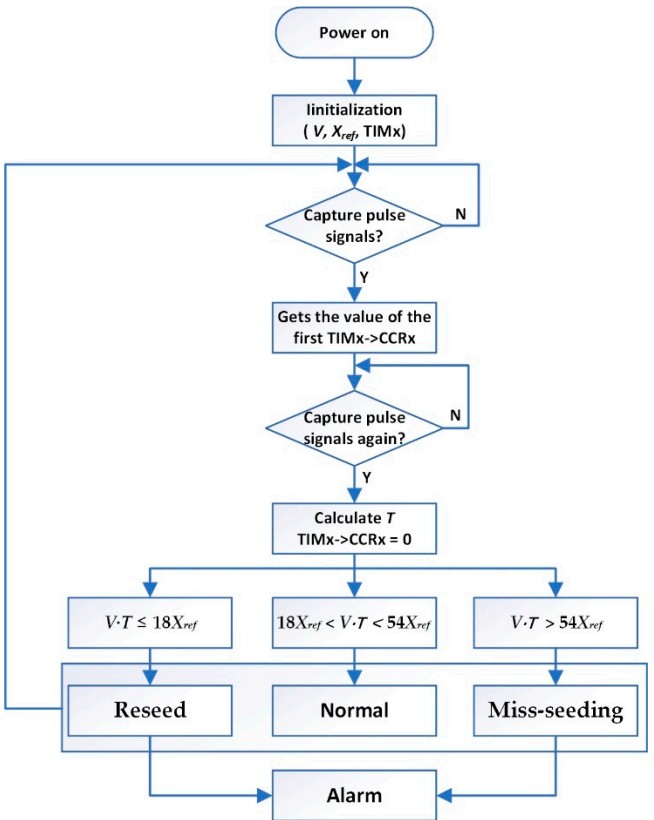

**Figure 10.** Seed condition monitoring process.

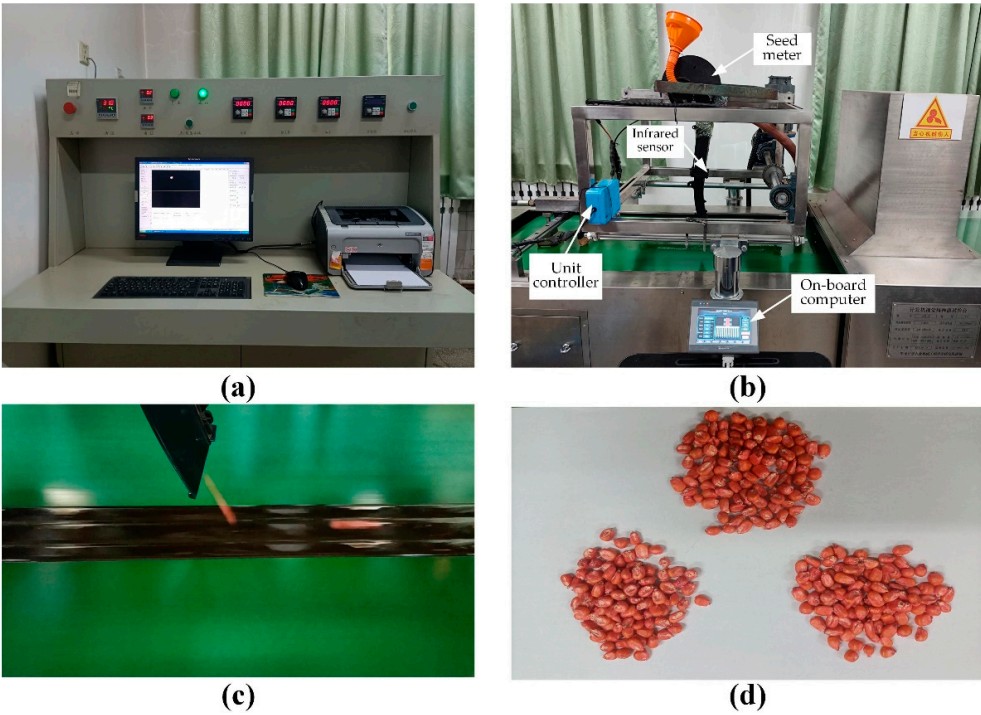

**Figure 11.** Seeding parameter monitoring on the JPS-12. (**a**) Control cabinet; (**b**) test bench; (**c**) seeding; (**d**) data statistics.

Since no interface can obtain the real-time speed on the JPS-12 test bench, to obtain the real-time operating speed of the seedbed belt as much as possible to simulate the field environment, ten groups of magnetic steel were installed on the inner side of the seedbed drive roller. NPN constant open all-pole Hall sensors were used in pulse signal detection. Figure 12 shows the installation position of the magnetic steel and the Hall sensors. The dynamic speed of the seedbed could be calculated according to Formula (3) after the signal of the speed pulse was collected by the Hall sensor.

$$V_b = \frac{\pi d n}{m T_c} \tag{3}$$

where $V_b$ is the speed of the seedbed belt (m/s), $d$ is the roller diameter (mm), $n$ is the number of pulses in the $T_c$ cycle, $m$ is the number of magnetic steels, and $T_c$ is the count cycle (ms).

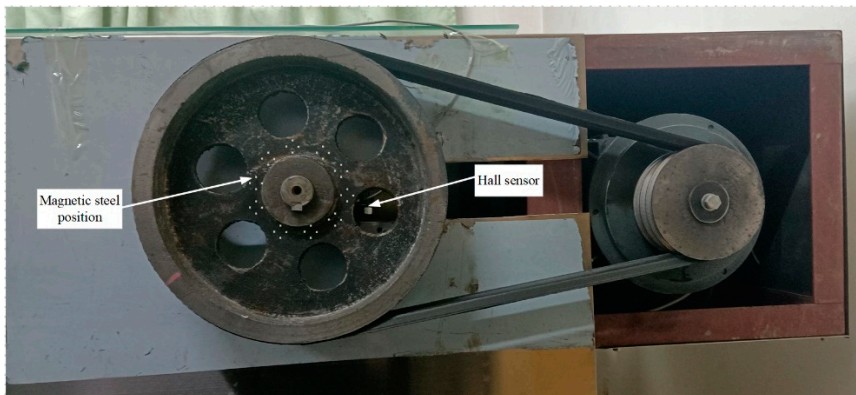

**Figure 12.** Schematic diagram of seedbed belt speed detection.

The field experiment was conducted in Xiangshui County, Yancheng City, Jiangsu Province, on 17 February 2022, using a dual row with an eighteen-row seeder developed by Devo, Heilongjiang Province (Figure 13). To explore the influence of different operating speeds on seeding performance, the negative pressure of the fan output was adjusted to 4.5 kPa, the grain spacing was set to 20 cm, and the operating speeds were changed to 8 km/h, 10 km/h, and 12 km/h. To explore the effects of different grain spacings on sowing performance, the operating speed was 8 km/h, and the grain spacings were changed to 15 cm, 20 cm, and 25 cm. At the same time, we explored the differences in sowing performance parameters between different planting units. The grain spacing data were obtained by manual measurement.

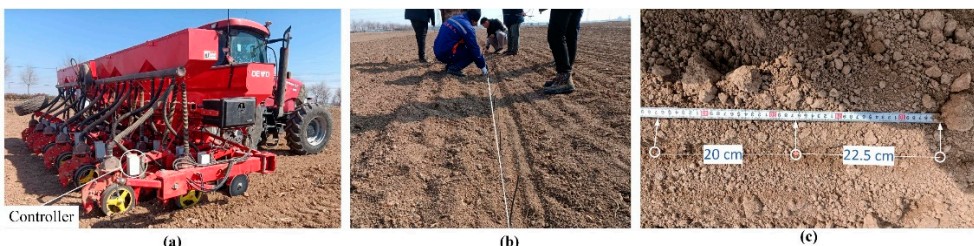

**Figure 13.** Precision electric seeder and monitoring system test. (**a**) Eighteen-row maize precision electric seeder and monitoring system; (**b**) label seed position; (**c**) seed spacing measurement.

According to GB/T 6973-2005, the qualified index, QI, reseed index, RI, missing index, MI, and coefficient of variation, CV, were calculated as evaluation indices of sowing quality.

$$QI = \frac{n_1}{N'} \times 100\% \tag{4}$$

$$\text{RI} = \frac{n_2}{N'} \times 100\% \tag{5}$$

$$\text{MI} = \frac{n_0}{N'} \times 100\% \tag{6}$$

$$X = \frac{\sum(n_i X_i)}{n_2} \tag{7}$$

$$\sigma = \sqrt{\frac{\sum(n_i X_i)^2}{n_2} - X^2} \tag{8}$$

$$\text{CV} = \sigma \times 100\% \tag{9}$$

where $N'$ is the total number of normalized intervals, $n_0$, $n_1$, and $n_2$ are the missing numbers ($X_i \in (1.5, +\infty]$), the qualified number ($X_i \in (0.5, 1.5]$), and the replay number ($X_i \in [0, 0.5]$), respectively, $n_i$ and $X_i$ are the grain spacing number and interval median in the $i$(th) interval, respectively, and $X$ and $\sigma$ are the mean and standard deviation of the sample, respectively. At the same time, these indicators were evaluated according to the NY/T 1143-2006 standard provided by the Ministry of Agriculture of China. Table 2 shows the main performance indices of the precision seeder.

**Table 2.** Main performance indices of the precision seeder.

| Index | Indicators | | |
| --- | --- | --- | --- |
| | Seed Spacing ≤10 cm | Seed Spacing >10 cm~20 cm | Seed Spacing >20 cm~30 cm |
| Qualified index | ≥60.0 | ≥75.0 | ≥80.0 |
| Reseeding index | ≤30.0 | ≤20.0 | ≤15.0 |
| Missing index | ≤15.0 | ≤10.0 | ≤8.0 |
| Coefficient of variation | ≤40.0 | ≤35.0 | ≤30.0 |

## 3. Results and Discussion

### 3.1. Photoelectric Sensor Monitoring Performance and Real-Time Online Monitoring Test

To test the performance of the photoelectric sensor, the numbers of monitored corn grains at speeds of 6, 8, 10, and 12 km/h were tested in the laboratory and the field. When the speed reached the set value, the test seeder monomer was started by the virtual button on the on-board computer, and the seeder monomer was stopped at a random time. The number of corn seeds collected in containers fixed below the metering tube was manually counted. The statistical results indicate that the photoelectric sensor monitoring performance was quite good, and there were no differences between the laboratory and field monitoring data. Table 3 shows the statistical results of the monitoring data and actual data. The average monitoring accuracy was 99.8%.

To test the reliability of the system fault alarm, two kilograms of corn seeds were added to each sowing monomer. In the initial stage of operation, the metering tube was in normal planting, and the system did not send alarm information. When the seed box was empty, the system was checked to determine whether the alarm was prompted and whether the corresponding sowing monomer was shown in the vehicle terminal. According to the same method, during the normal seeding period, the seeding tube was artificially blocked at a given time, and the system blocking alarm rate was checked. The test results of fifty trials showed that the fault alarm rate was 100%.

The statistical analysis of the performance index data is shown in Figure 14. It can be seen from the chart that each evaluation index was basically similar at different speeds and substantially exceeded the standard (NY/T 1143-2006).

Overall, the qualified rate was higher when the grain spacing was larger. It was also found that when the speed was 12 km/h, the qualified rate decreased compared with the other speeds and the missed rate increased. The reason is that with the increase in the speed of the seedbed belt, the sliding degree of the seedbed pulley relative to the seedbed

belt increased, resulting in inaccurate speed measurement. When the speed was 8 km/h and 10 km/h, the consistency of the indices is good, and the difference was significant when the speed was 6 km/h and 12 km/h.

**Table 3.** Statistical results of monitoring data and actual data.

| Site | Speed (km/h) | Monitoring Value | | | Actual Value | | |
|---|---|---|---|---|---|---|---|
| | | $A_m$ | $B_m$ | $C_m$ | $A_a$ | $B_a$ | $C_a$ |
| Laboratory | 6 | 245 | 304 | 258 | 245 | 305 | 258 |
| | 8 | 275 | 236 | 459 | 275 | 236 | 460 |
| | 10 | 236 | 389 | 321 | 236 | 389 | 321 |
| | 12 | 335 | 286 | 462 | 335 | 287 | 462 |
| Field | 6 | 365 | 428 | 303 | 365 | 428 | 303 |
| | 8 | 274 | 354 | 482 | 275 | 355 | 482 |
| | 10 | 382 | 298 | 407 | 382 | 299 | 408 |
| | 12 | 473 | 387 | 496 | 473 | 385 | 496 |

Table 4 shows the correlation analysis between the factors and performance indicators. There was a strong correlation between two factors (*V* and $X_{ref}$) and the seed distribution uniformity index (CV, QI, RI, and MI). In addition, the statistical values describing the correlation between various factors and performance indicators show that there was a strong correlation between QI, RI, MI, and *V*: QI decreased with an increase in *V* and RI, and MI increased with an increase in *V*; there were significant correlations between QI and CV and between RI and MI. The CV, RI, and MI decreased with increasing QI, and the correlation between MI and QI was the strongest. The coefficient of determination was 0.983, and the level of visibility was far less than 0.01.

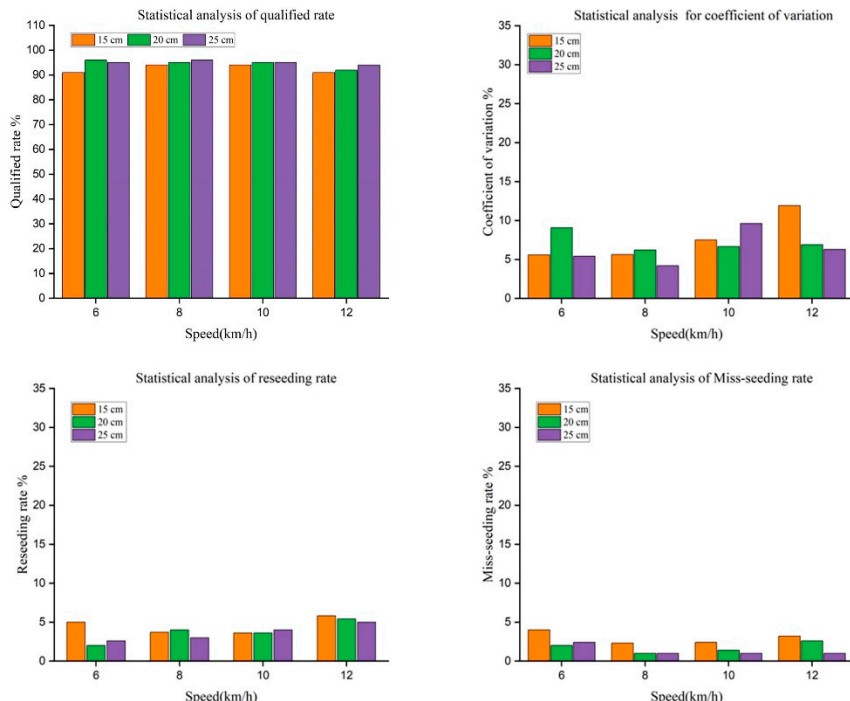

**Figure 14.** Statistical analysis of the performance index data.

**Table 4.** Correlation analysis between factors and performance indicators.

| Items | $V$/(km/h) | $X_{ref}$/mm | CV/% | QI/% | RI/% | MI/% |
|---|---|---|---|---|---|---|
| $V$/(km/h) | 1.000 [a] \ [b] | 0.000 1.000 | 0.527 0.145 | −0.791 * 0.011 | 0.738 * 0.023 | 0.843 ** 0.004 |
| $X_{ref}$/mm | 0.000 1.000 | 1.000 \ | −0.369 0.329 | 0.527 0.145 | −0.264 0.493 | −0.474 0.197 |
| CV/% | 0.527 0.145 | −0.369 0.329 | 1.000 \ | −0.733 * 0.025 | 0.717 * 0.030 | 0.633 0.067 |
| QI/% | −0.791 * 0.011 | 0.527 0.145 | −0.733 * 0.025 | 1.000 \ | −0.867 ** 0.002 | −0.983 ** 0.000 |
| RI% | 0.738 * 0.023 | −0.264 0.493 | 0.717 * 0.030 | −0.867 ** 0.002 | 1.000 \ | 0.833 ** 0.005 |
| MI/% | 0.843 ** 0.004 | −0.474 0.197 | 0.633 0.067 | −0.983 ** 0.000 | 0.833 ** 0.005 | 1.000 \ |

Note: Because the prior uncertainty is a positive correlation or negative correlation, the double tail test was chosen; descriptive statistics of sample data were used to calculate the average and variance; and the visibility of the output results must be marked. When the visibility level reaches 0.05, the upper right corner uses '*'; when the visibility level reaches 0.01, the upper right corner uses '**'. [a] represents the coefficient of determination; [b] represents the *p* value, namely, the level of dominance.

### 3.2. Differences in Seeding Performance among Different Planting Units in the Field

The test results of the seeding performance at different operating speeds are shown in Table 5. When the operating speed was 8 km/h, the seeding performance was excellent. The qualified index of single seeding was 94.14%, the reseeding index was 1.72%, and the missing seeding index was 4.14%. With the increase in the operation speed, the reseeding index always maintained a certain level. However, due to the insufficient wind pressure of the fan and the irregular bounce of the seed when landing, the missing seeding index increased significantly, resulting in a decrease in the seeding accuracy. When the operating speed was 10 km/h, the seeding qualified index was reduced to 91.48%, and the leakage index was increased to 7.46%. When the operating speed was 12 km/h, the seeding qualified index was still greater than 90%.

**Table 5.** Results of the seeding performance at different operating speeds.

| Items | $V = 6$ (km/h) | | $V = 8$ (km/h) | | $V = 10$ (km/h) | | $V = 12$ (km/h) | |
|---|---|---|---|---|---|---|---|---|
| | No. 2 | No. 7 | No. 2 | No. 7 | No. 2 | No. 7 | No. 2 | No. 7 |
| Average distance (cm) | 19.38 | 19.24 | 19.10 | 19.40 | 20.30 | 19.18 | 21.50 | 20.31 |
| QI/% | 93.17 | 93.40 | 94.14 | 94.53 | 91.48 | 91.94 | 90.35 | 90.01 |
| RI/% | 1.38 | 1.86 | 1.72 | 1.14 | 1.06 | 0.80 | 1.02 | 1.93 |
| MI/% | 5.45 | 4.74 | 4.14 | 4.33 | 7.46 | 7.26 | 8.63 | 8.06 |
| Standard deviation | 5.32 | 6.73 | 4.59 | 5.46 | 9.65 | 8.41 | 10.01 | 9.35 |

Note: No. 2 and No. 7 represent the second and seventh sowing planting units, respectively.

The seeding performance test results at different seed spacing settings are shown in Table 6, and the operating speed remained 8 km/h. With the increase in the seed spacing, the qualified index decreased and the reseeding index and leakage index increased. For a grain spacing of 15 cm, the average qualified index of two single seedlings was 93.34%, the average reseeding index was 2.09%, and the average missing seeding index was 4.57%.

When the operating speed was within the range of 6~12 km/h or the grain spacing was set to 15~25 cm, there was no significant difference in the seeding performance between the No. 2 and No. 7 planting units. These differences may be caused by factors such as the processing technology of the planting unit mechanical mechanism, mechanical vibration, and measurement error. Therefore, it is considered that the variability of the seeding performance between the monomers is small. In summary, when the quality of seeds and soil preparation meets the agronomic requirements of sowing, the electric drive seeding

control system designed in this study meets the requirements of precision sowing under high-speed working conditions.

**Table 6.** Seeding performance for different driving modes.

| Items | $X_{ref}$ = 15 cm | | $X_{ref}$ = 20 cm | | $X_{ref}$ = 25 cm | |
|---|---|---|---|---|---|---|
| | No. 2 | No. 7 | No. 2 | No. 7 | No. 2 | No. 7 |
| Average distance (cm) | 14.43 | 14.15 | 19.93 | 19.28 | 23.10 | 23.40 |
| QI/% | 91.79 | 92.25 | 93.19 | 93.98 | 94.12 | 94.71 |
| RI/% | 2.65 | 3.01 | 2.12 | 1.89 | 1.72 | 1.14 |
| MI/% | 5.56 | 4.74 | 4.69 | 4.13 | 4.16 | 4.15 |
| Standard deviation | 7.61 | 8.26 | 6.32 | 5.78 | 4.59 | 5.46 |

Note: No. 2 and No. 7 represent the second and seventh sowing planting units, respectively.

*3.3. Discussion of the Results*

Based on the control system of the electric drive precision seeder, laboratory bench tests and field tests were carried out. Its performance indicators tended to be consistent, which also fully illustrated the system reliability. The bench test explored the effects of different operating speeds and grain spacing on the seeding performance indices. At present, many scholars have carried out electric drive seeding experiments, and their working performance has been greatly improved compared with the traditional mechanical seeders. The performance indicators involved in this study are similar to those used in previous studies. Due to the differences in the environment and mechanical structure, the qualified rate of sowing in the field was lower than that of the bench test.

Since the test bench is designed for a traditional mechanical seeder, the influence of the seedbed vibration and slip ratio of the seedbed belt during high-speed operation has not been fully considered, thus affecting the test results to a certain extent [33]. In the field experiment, previous researchers mostly used 4-row or 6-row seeding machines for experiments. In this study, an 18-row air suction precision seeding machine was used. Due to the increase in seeding monomers, the airflow of the fan was unstable at high speeds, resulting in insufficient pressure during high-speed operation and a slight decrease in the seeding qualified index; seeding monomers on both sides of the seeding machine was a common malfunction. Nevertheless, more than 90% of the qualified rates fully met the actual work requirements. In the selection of photoelectric sensors, based on previous studies, a rectangular infrared radiation surface was selected, which greatly improved the sensing area of the photoelectric sensors and reduced the blind area. The high sensitivity of the sensor increased the fault alarm rate.

The sensors used in the system and the electronic components used in the design circuit are commonly used in the market. Compared with the laser detection sensor used in [7], the photoelectric sensor has a high value for practical application, and the monitoring performance was better than the laser detection performance; compared with the expensive LiDAR used in [10], the system used the common satellite acquisition module and achieved good data acquisition and control effect through certain filtering algorithms. The CAN bus control method greatly reduces the difficulty of field wiring. The brush DC motor is easier to control and lower cost than the brushless DC motor used in [17,18]. Usually, brushless motors perform better than brush motors.

In order to prevent electrostatic interference to the system, the electrostatic shielding circuit was specially designed in the circuit, which improved the anti-interference properties and robustness of the system. In practical field applications, a shielded twisted pair is used in CAN bus transmission, and terminal resistance is connected to the transceiver end. At the same time, CAN bus through the data link layer and physical layer has achieved high bus data security and bus stability; the correctness of data transmission is ensured by establishing a CANopen object dictionary. The above measures enhance the robustness of the system to subsystem faults and electromagnetic interference. Overall,

whether in economic cost or system performance, the system was suitable for agricultural machinery operation.

On the other hand, the acquisition accuracy of the tractor speed directly affects the sowing quality. Although the accuracy of GPS can meet the current operating requirements, once the GPS signal, as the only acquisition speed, is affected, it seriously affects the operating quality. In the future, multisensor information fusion technology will be used to compensate for the speed signal to ensure that the speed measurement accuracy can still be maintained under sensor fault and interference conditions to ensure the consistency of the operation quality in a complex working environment.

## 4. Conclusions

A control system of an electrically driven precision maize seeder based on the CANopen protocol was designed and developed. A circuit board with motor drive and sowing performance detection was integrated. The matching model of vehicle speed and seeding plate speed was established through the PID control algorithm. Terminal monitoring software for real-time monitoring of sowing parameters was designed. According to the GB/T 6973-2005 standard, the evaluated parameters were the following: photoelectric sensor detection performance, fault alarm rate, qualified rate, reseeding rate, and missed rate. The following conclusions can be drawn:

(1) In terms of photoelectric sensor detection performance, there was not a large difference between the indoor bench tests and field tests with dust pollution, and the detection accuracy reached 99.8%. This also shows that the sensor has a strong penetration ability and a large radiation detection surface. The fault alarm function of the system was accurate and timely, and the fault alarm rate reached 100%.

(2) Based on the indoor test results, the qualified rate was higher when the grain spacing was larger. It was also found that when the speed was 12 km/h, the qualified rate decreased compared with other speeds, and the missed rate increased. When the speed was 8 km/h and 10 km/h, the consistency of the indices was good, and the difference was significant when the speed was 6 km/h and 12 km/h. Overall, the qualified rate of sowing was more than 91%. At the same time, the correlation of the seeder index parameters was analyzed: there were strong correlations between QI, RI, MI, and $V$; QI decreased with increasing $V$; RI and MI increased with increasing $V$; and CV, RI, and MI decreased with increasing QI. Furthermore, the correlation between MI and QI was the strongest. The coefficient of determination was 0.983, and the level of visibility was far less than 0.01.

(3) Based on field test results, the seeding performance results showed that the control system has good stability. When the grain spacing was set to 20 cm and the operating speed was 6~12 km/h, the qualified index was more than 90%, and the reseeding index was less than 1.93%. The variation in sowing performance between different monomers was small, and the seeding performance was good, which can provide a reference for the development and design of high-speed precision corn seeders.

**Author Contributions:** Conceptualization, J.C., C.J. and M.D.; methodology, J.C. and F.P.; software, J.C.; validation, J.C., C.J. and H.Z.; formal analysis, J.C. and M.D.; investigation, J.C., C.J. and H.Z.; resources, C.J. and M.D.; data curation, J.C. and C.J.; writing—original draft preparation, J.C.; writing—review and editing, J.C. and C.J.; visualization, J.C. and H.Z.; supervision, C.J.; project administration, C.J. and M.D.; funding acquisition, C.J. and M.D. All authors have read and agreed to the published version of the manuscript.

**Funding:** This research was funded by the Key Scientific and Technological Projects in Key Areas of the Xinjiang Production and Construction Corps (No. 2020AB011) and the National Natural Science Foundation of China (NSFC) (No. 32101634).

**Institutional Review Board Statement:** Not applicable.

**Informed Consent Statement:** Not applicable.

**Data Availability Statement:** Data are contained within the article.

**Conflicts of Interest:** The authors declare no conflict of interest.

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
