# Peer review of "Control System of a Motor-Driven Precision No-Tillage Maize Planter Based on the CANopen Protocol"

_agriculture, doi:10.3390/agriculture12070932_

Round 1

Reviewer 1 Report

The paper proposes a control system of a motor-driven precision no-tillage maize planter based on the CANopen protocol. 

The proposed solution is compact and it presents interesting results for indoor and field tests.

I recomend the authors to present a comparative in terms of costs and, if possible in this stage of research, to write something about the robustness of such solution for real application in field.

The paper addresses an important research topic and it is interesting to read.

Author Response

Point 1: I recommend the authors to present a comparative in terms of costs and, if possible in this stage of research, to write something about the robustness of such solution for real application in field.

Response 1: A paragraph has been added to section 3.3 (Discussion of the Results) for discussion and comparison in terms of cost(line163-471).

For the system robustness problem proposed by experts, A paragraph has been added to section 3.3 (Discussion of the Results) for discussion(line472-480).

Reviewer 2 Report

The paper is well written. You have described a modification to a maize planter that is intended to address concerns with drive-wheel mechanisms which tend to be unreliable as field velocity increases. The prototype was tested in both the laboratory environment and the field environment. Appropriate data were collected to evaluate the performance of the motor-driven control system. Clearly stated conclusions have been provided. Overall, the paper is well written. I do not feel qualified to comment on the description of the electronics.

Author Response

Point 1: The paper is well written. You have described a modification to a maize planter that is intended to address concerns with drive-wheel mechanisms which tend to be unreliable as field velocity increases. The prototype was tested in both the laboratory environment and the field environment. Appropriate data were collected to evaluate the performance of the motor-driven control system. Clearly stated conclusions have been provided. Overall, the paper is well written. I do not feel qualified to comment on the description of the electronics.

Response 1: We would like to express our great appreciation to you and reviewers for comments on our paper, which is also an encouragement to us.

Reviewer 3 Report

Your manuscript is suitable for publication in agriculture. I recognize that the topic is important, the content, style and format needs work to be suited for “agriculture”. The manuscript entitled “Control System of a Motor-driven Precision no-tillage Maize Planter Based on the CANopen protocol” is an interesting topic. Authors need to be as specific as possible in the manuscript.

Here are my recommendations for revisions:

 (1) Concerning the English some minor changes must be made and it is recommended to the authors to proofread the manuscript by a native speaker.

(2) The control algorithm is too simple, might be some sort of advanced control technique can be used.

(3) Figure 11 can be replaced with a clear picture of the experimental setup. It is not necessary to show people who are performing experiments.

(4) To verify the effectiveness of the proposed work, a comparison is important with published literature work. 

Author Response

Point 1: Concerning the English some minor changes must be made and it is recommended to the authors to proofread the manuscript by a native speaker.

Response 1: Thank you for your suggestions. According to your advice, this manuscript was edited for proper English language, grammar, punctuation, spelling, and overall style by one or more of the highly qualified native English-speaking editors at AJE. 

Point 2: The control algorithm is too simple, might be some sort of advanced control technique can be used. 

Response 2: In terms of algorithm, we have tried intelligent control algorithms such as particle swarm optimization algorithm, adaptive control algorithm, and fuzzy control algorithm. In terms of performance improvement, it has not been greatly improved compared with the classical PID algorithm, and the response time is less than the PID control. Especially, due to the lack of computing resources of the control chip used in this system, the use of more complex control algorithm occupied a large number of CPU computing resources, which sometimes leads to other control process instability. In addition, in view of the actual working situation, the feedback time response of the signal has a great influence on the working quality, and the speed and voltage of the motor are close to a linear relationship, which is more suitable for PID control algorithm. According to the actual control situation, it fully meets the actual control requirements. For these reasons, more complex control algorithms are not applied yet in this system. Perhaps more complex algorithms can be applied to better controllers in the future. 

Point 3: Figure 11 can be replaced with a clear picture of the experimental setup. It is not necessary to show people who are performing experiments. 

Response3: Figure 11(d) has been replaced with other pictures without people who are performing experiments. 

Point4:To verify the effectiveness of the proposed work, a comparison is important with published literature work. 

Response 4: A paragraph has been added to section 3.3 (Discussion of the Results) for comparison with published literature work(line163-471).
